# The Use of Predictive Microbiology for the Prediction of the Shelf Life of Food Products

**DOI:** 10.3390/foods12244461

**Published:** 2023-12-13

**Authors:** Fatih Tarlak

**Affiliations:** Department of Nutrition and Dietetics, Faculty of Health Sciences, Istanbul Gedik University, Kartal, Istanbul 34876, Turkey; ftarlak@gtu.edu.tr; Tel.: +90-(216)-444-5438

**Keywords:** modelling, microbial growth, spoilage, machine learning approach

## Abstract

Microbial shelf life refers to the duration of time during which a food product remains safe for consumption in terms of its microbiological quality. Predictive microbiology is a field of science that focuses on using mathematical models and computational techniques to predict the growth, survival, and behaviour of microorganisms in food and other environments. This approach allows researchers, food producers, and regulatory bodies to assess the potential risks associated with microbial contamination and spoilage, enabling informed decisions to be made regarding food safety, quality, and shelf life. Two-step and one-step modelling approaches are modelling techniques with primary and secondary models being used, while the machine learning approach does not require using primary and secondary models for describing the quantitative behaviour of microorganisms, leading to the spoilage of food products. This comprehensive review delves into the various modelling techniques that have found applications in predictive food microbiology for estimating the shelf life of food products. By examining the strengths, limitations, and implications of the different approaches, this review provides an invaluable resource for researchers and practitioners seeking to enhance the accuracy and reliability of microbial shelf life predictions. Ultimately, a deeper understanding of these techniques promises to advance the domain of predictive food microbiology, fostering improved food safety practices, reduced waste, and heightened consumer confidence.

## 1. Introduction

The term “food shelf life” refers to the duration in which a food product sustains its safety, quality, and nutritional attributes within specified storage conditions. This interval encompasses the period during which the food item remains appropriate for consumption and retains its intended sensory, physical, chemical, and microbiological properties, aligning with the manufacturer’s intentions [1]. The assessment and determination of shelf life play a pivotal role in guaranteeing consumer well-being, minimising food waste, and ensuring economic sustainability.

Shelf life pertains to the time frame during which a food product maintains its intended quality and safety characteristics while subjected to specific storage conditions [2]. Its influencing factors comprise microbial growth, chemical reactions, physical alterations, and sensory degradation. To ensure consumer safety and satisfaction, the precise estimation of shelf life and the formulation of strategies for its extension are of the utmost importance.

In the realm of the food industry, food shelf life constitutes a fundamental cornerstone, embodying a nuanced equilibrium among safety, quality, and consumer contentment. The intricate interplay among sensory, physical, chemical, and microbiological elements necessitates a rigorous scientific approach and comprehensive comprehension [3]. Through an accurate determination and the proficient management of shelf life, stakeholders within the food supply chain can uphold consumer safety, mitigate wastage, and maintain economic feasibility, thereby contributing to a sustainable and responsible food sector.

Food shelf life pertains to the timeframe in which a food product retains its intended quality, safety, and nutritional characteristics under specific storage conditions. This duration represents the period during which the food item remains suitable for consumption while retaining its desired sensory, physical, chemical, and microbiological attributes, as intended by the manufacturer or producer. The process of determining food shelf life involves evaluating factors such as microbial growth, chemical reactions, physical changes, and sensory deterioration, ensuring the product’s ongoing safety and satisfactory attributes for consumers. The accurate determination and effective management of shelf life are imperative to ensure consumer safety, minimise food waste, and sustain economic viability within the context of the food industry. Several factors influence the shelf life of food products [4,5,6]:➢**Microbial Activity:** Microorganisms, including bacteria, yeasts, and moulds, play significant roles in food spoilage and degradation. Their growth can lead to changes in flavour, texture, odour, and overall quality [7]. ➢**Chemical Reactions:** Chemical reactions such as oxidation, enzymatic reactions, and hydrolysis can cause changes in colour, taste, nutritional content, and texture. These reactions are often accelerated by factors like temperature, light, and oxygen exposure [8].➢**Physical Changes:** Physical changes like moisture migration, crystallisation, and phase separation can affect the appearance, texture, and stability of food products [9]. ➢**Water Activity (Aw):** Water activity refers to the amount of available water in a product. Microbial growth and chemical reactions are often inhibited at lower water activity levels [10].➢**Temperature:** Temperature is a critical factor influencing shelf life. Higher temperatures can accelerate chemical reactions and microbial growth, leading to faster deterioration [11].➢**Packaging:** Packaging materials and methods can impact the shelf life of a product by influencing factors such as oxygen and moisture permeability [12]. ➢**pH:** The pH level of a food product can affect microbial growth and enzyme activity. Acidic environments can inhibit the growth of spoilage organisms [13].➢**Preservatives:** The addition of preservatives like antioxidants, antimicrobials, and flavour enhancers can extend shelf life by inhibiting microbial growth and delaying oxidation [14]. ➢**Storage Conditions:** Storage conditions, including temperature, humidity, and exposure to light, significantly impact the rate of deterioration. Proper storage is essential to maintaining product quality [15]. 

### Microbial Shelf Life

Microbial shelf life refers to the duration of time during which a food product remains safe for consumption in terms of its microbiological quality. It is the period during which the microbial population within the product remains within acceptable limits, ensuring that the food item is free from harmful microorganisms that can cause spoilage or pose health risks to consumers [16]. The determination of microbial shelf life involves monitoring the growth and activity of microorganisms, such as bacteria, yeasts, and moulds, to ensure that their populations remain controlled and do not reach levels that compromise the safety or quality of the food product. An accurate estimation of microbial shelf life is crucial for ensuring the safety and freshness of foods and preventing the onset of microbial-related deterioration.

Microbial shelf life is influenced by a variety of factors that impact the growth and activity of microorganisms within a food product [17]. These factors determine how long a product can remain safe for consumption before microbial populations reach levels that compromise its quality and safety. Some of the key factors affecting microbial shelf life include:

Microbial growth is influenced by a variety of factors that determine how quickly microorganisms multiply and proliferate within a given environment. These factors play a crucial role in determining the safety, quality, and shelf life of food products. Some of the key factors affecting microbial growth include:➢**Nutrients:** Microorganisms require nutrients such as carbohydrates, proteins, and fats for growth. Foods rich in these nutrients can provide an environment conducive to microbial proliferation [18]. ➢**Water Activity (Aw):** Water activity refers to the availability of water for microbial growth. Microorganisms require water to carry out metabolic processes. Foods with higher water activity levels offer more favourable conditions for microbial growth [19].➢**Temperature:** Temperature has a profound impact on microbial growth rates. The relationship between temperature and microbial growth is often described by the “temperature danger zone”, within which microorganisms multiply most rapidly. Cold temperatures slow down microbial growth, while temperatures above the danger zone can kill some microorganisms [20].➢**pH:** Microorganisms have specific pH ranges in which they thrive. Bacteria generally prefer neutral pH conditions, while moulds and yeasts can tolerate a wider pH range. Extreme pH values can inhibit microbial growth [21]. ➢**Oxygen Availability:** Microorganisms can be classified into aerobic (requiring oxygen), anaerobic (thriving in the absence of oxygen), and facultative aerobe (growing in presence or absence of oxygen) categories [22]. Oxygen availability influences the types of microorganisms that can grow and the rate of their growth [23].➢**Redox Potential:** Redox potential measures the availability of electrons in an environment. It affects the growth of both aerobic and anaerobic microorganisms [19].➢**Presence of Antimicrobial Compounds:** Some foods naturally contain compounds with antimicrobial properties, such as spices, herbs, and essential oils. These compounds can inhibit or slow microbial growth [24].➢**Surface Area:** Larger surface areas provide more opportunities for microorganisms to attach and grow. Cutting or grinding food increases its surface area, potentially promoting microbial growth.➢**Moisture Content:** The moisture content of a food product affects its water activity and can impact microbial growth. High-moisture foods are generally more prone to microbial proliferation [25].➢**Intrinsic Factors:** Intrinsic factors are inherent characteristics of the food itself, such as its composition, structure, and natural microflora. These factors can influence the types of microorganisms that grow and the rate at which they do so.

By understanding these factors and how they interact, food producers and scientists can develop strategies for controlling and managing microbial growth, ensuring the safety and quality of food products throughout their shelf life. A sensory analysis assesses food attributes like taste and texture, but it does not directly improve microbial shelf life predictions. These predictions rely on specific data about microorganisms and food spoilage. However, combining a sensory analysis with microbial data can offer a full picture of product stability, covering safety and taste. A sensory analysis alone is not enough; it should work with microbial data, environmental monitoring, and advanced modelling for a complete quality and safety assessment.

## 2. Predictive Microbiology

Predictive microbiology is a field of science that focuses on using mathematical models and computational techniques to predict the growth, survival, and behaviour of microorganisms in food and other environments [26]. This approach allows researchers, food producers, and regulatory bodies to assess the potential risks associated with microbial contamination and spoilage, enabling informed decisions to be made regarding food safety, quality, and shelf life.

➢**Microbial Growth Models:** Predictive microbiology often involves the development of mathematical models that describe the growth of microorganisms under specific conditions. These models take into account factors such as temperature, pH, water activity, and initial microbial load to estimate the rate and extent of microbial proliferation.➢**Risk Assessment:** Predictive models are used to assess the potential risks of microbial contamination in food products. By simulating different scenarios, regarding the various crucial factors that influence microbial contamination in food products such as the specific environmental conditions [27,28], intrinsic and extrinsic parameters of the food [29], processing techniques, and storage conditions [30,31], researchers can determine how different environmental conditions impact the growth of pathogenic and spoilage microorganisms.➢**Shelf Life Estimation:** Predictive microbiology helps in estimating the shelf life of food products. By considering microbial growth rates and spoilage thresholds, manufacturers can determine how long a product can remain safe and of an acceptable quality under various storage conditions.➢**Quality Control:** Predictive models aid in establishing critical control points in the production process where microbial growth can be controlled or prevented. This supports quality assurance and helps to prevent foodborne illnesses.➢**Regulatory Compliance:** Regulatory bodies often rely on predictive microbiology models to set standards and guidelines for food safety. These models provide insights into safe storage conditions and acceptable microbial levels.➢**Advancements in Technology:** With the integration of data science and advanced computational tools, predictive microbiology has evolved to include machine learning and artificial intelligence algorithms that can analyse complex datasets to enhance predictions.

Predictive microbiology plays a pivotal role in enhancing our understanding of microbial behaviour, food safety, and quality control. By leveraging mathematical models and computational tools, this field empowers the food industry to make data-driven decisions that ensure consumer safety and satisfaction. The application of predictive microbiology has greatly enhanced the food industry’s capacity to forecast and regulate microbial shelf life, leading to advancements in food safety and quality. Through the utilisation of mathematical models and computational methods, researchers have gained valuable insights into the dynamics of microorganism growth and behaviour of various food products [32]. These predictive models enable the evaluation of potential risks linked to microbial contamination, facilitating informed decisions concerning food preservation, storage, and distribution [33]. Moreover, the implementation of predictive microbiology has facilitated the formulation of effective approaches to prolonging the shelf life of food products, thereby reducing food waste and ensuring consumer health and satisfaction [34].

### 2.1. Primary Models

The modified Gompertz, logistic, Baranyi, and Huang models, in particular, are the most often utilised sigmoid functions for describing bacterial growth behaviour. Equations (1) and (2), respectively, define the modified Gompertz and logistic models under constant environmental circumstances [35]:(1)$${x\left(t\right)=x_{0}+(x}_{max}-x_{0}).exp\left\{-exp\left[\frac{r_{max}.e}{{(x}_{max}-x_{0})}.\left(\lambda -t\right)+1\right]\right\}$$
(2)$$x\left(t\right)=x_{0}+\frac{{(x}_{max}-x_{0})}{\left\{1+\exp\left[\frac{{4.r}_{max}}{{(x}_{max}-x_{0})}.\left(\lambda -t\right)+2\right]\right\}}$$
where t is the time (h), x(t) is the bacterial population concentration (log CFU/g) at time t, x_0_ is the initial bacterial population concentration (log CFU/g), x_max_ is the maximum bacterial population concentration (log CFU/g), r_max_ is the maximum bacterial growth rate (log CFU/h), and λ is the lag phase duration (h).

Other widely used primary functions include the Baranyi and Huang models, which are represented by Equations (3) and (4), respectively [36,37]:(3)$$y\left(t\right)=y_{0}+{µ}_{max}F\left(t\right)-ln\left(1+\frac{e^{{µ}_{max}F\left(t\right)}-1}{e^{\left(y_{max}-y_{0}\right)}}\right)$$
(4)$$y\left(t\right)=y_{0}+y_{max}-ln(e^{y_{0}}+\left[e^{y_{max}}-e^{y_{0}}\right].e^{{-µ}_{max}B(t)}$$
where t is the time (h), y(t) is the bacterial population concentration (ln CFU/g) at time t, y_0_ is the initial bacterial population concentration (ln CFU/g), y_max_ is the maximum bacterial population concentration (ln CFU/g), µ_max_ is the maximum specific bacterial growth rate (1/h), λ is the lag phase duration (h), and F(t) and B(t) are the adjustment functions described [36,37].

Because the major models utilise different scales for counting microbe populations, after fitting, the growth rate values (r_max_) derived from the Modified Gompertz and logistic models are translated into maximum specific growth rate values (µ_max_) by multiplying ln [38].

### 2.2. Secondary Models

Secondary models are used to describe the impacts of many environmental conditions on the parameters of main models, such as water activity, acidity, and temperature [38]. These secondary models are typically applied subsequent to fitting the growth data to primary models. Understanding the impact of water activity [39] acidity [39], and temperature on growth rate is critical for effectively managing food preservation and ensuring product safety and quality [39,40,41,42]. The Ratkowsky model is used to explain the link between temperature and maximal specific growth rate [43]. The Arrhenius model is frequently used to characterise the influence of storage temperature on microbial development in foods and is commonly used to describe the temperature dependency of chemical processes [44]. To characterise the influence of water activity, acidity, and temperature on the maximum specific growth rate and lag phase duration, (Equations (5)–(9)) models are widely used:(5)$$\mu_{max}=b_{1}\left(a_{w}-a_{w\_min}\right)$$
(6)$$\mu_{max}=b_{1}\left(pH-{PH}_{min}\right).\left(pH-{PH}_{max}\right)$$
(7)$${\mu_{max}=b}_{1}{\left(T-T_{0}\right)}^{2}$$
(8)$${\mu_{max}=b}_{1}\left[exp\left(\frac{{-E}_{a}}{R\theta }\right)\right]$$
(9)$$\mu_{max}=\frac{1}{\lambda }$$
where µ_max_ is the maximum specific growth rate (1/h) obtained from the primary model, a_w_ is the water activity, and a_w_min_ is the minimal water activity at which growth stops. pH_min_ is the minimal pH and pH_max_ is the maximal pH at which growth stops. T is the temperature (°C), T_0_ is the theoretical minimum temperature (°C) for microbial growth, λ is the lag phase duration (h) collected from the primary model, b_1_ is the regression coefficient, E_a_ is the activation energy (J/mol), R is the universal gas constant (8.314 J/mol K), and θ is the absolute temperature (K).

### 2.3. Comparison of the Goodness of Fit of the Models

The root mean square error (RMSE), adjusted coefficient of determination (R^2^_adj_), Akaike information criteria (AIC), and Bayesian information criterion (BIC) may be used to compare the models’ estimate skills using Equations (10)–(13) correspondingly [45]:(10)$$RMSE=\sqrt{\sum_{i=1}^{n}\frac{{\left(x_{obs}-x_{fit}\right)}^{2}}{n-s}}$$
(11)$$R_{adj}^{2}=1-\left(\frac{n-1}{n-s}\right)\left(\frac{SSE}{SST}\right)$$
(12)$$AIC=n\;ln\left(\frac{SSE}{n}\right)+2s$$
(13)$$BIC=n\;ln\left(\frac{SSE}{n}\right)+s\;ln(n)$$
where x_obs_ represents the experimental bacterial growth concentration, x_fit_ represents the fitted value, n represents the number of experiments, s represents the number of model parameters, SSE represents the sum of squares of errors, and SST represents the total sum of squares.

### 2.4. The Models’ Validation

The validation of models is the process through which the predictive power of the constructed models is validated using previously published or newly generated data. The ability of the models to forecast may be assessed using the microbes’ growth kinetics. Each of the global models’ related bias (B_f_) and accuracy (A_f_) factors are presented in Equations (14) and (15), respectively, for comparison [46]:(14)$$B_{f}=10^{\frac{\sum_{i=1}^{n}\log\left(\frac{x_{pred}}{x_{obs}}\right)}{n}}$$
(15)$$A_{f}=10^{\frac{\sum_{i=1}^{n}\left|log(x_{pred}/x_{obs})\right|}{n}}$$
where x_pred_ denotes the projected maximum values (1/h) and (h), x_obs_ denotes the experimental µ_max_ (1/h) and λ (h), and n is the number of experimental growth data.

## 3. Two-Step Modelling Approach

A two-step modelling approach, often referred to as a two-stage or dual-stage modelling approach, involves using two separate modelling techniques or stages to analyse a complex problem or dataset. Each stage serves a specific purpose and builds upon the results or insights from the previous stage [47]. This approach is commonly used when a problem is too intricate to be solved with a single model or when different aspects of the problem require different modelling techniques.


**Step 1: First-stage Modelling**


In the first stage, a preliminary model is developed to address a specific aspect of the problem [48]. This model is usually simpler and aims to provide initial insights or predictions. The output of this stage is used as the input for the second stage. For example, a basic microbial growth model might be used to predict the growth of a specific microorganism under varying temperature conditions. This provides an initial understanding of the relationship between temperature and microbial growth.


**Step 2: Second-stage Modelling**


In the second stage, a more complex or refined model is developed to incorporate additional factors or address other aspects of the problem. The output from the first stage serves as the input for the second-stage model. This stage typically provides more detailed and accurate predictions or insights. The output from the first-stage model (microbial growth under varying temperatures) can be used as the input for a more comprehensive model that considers other variables such as pH, water activity, and initial microbial load [38]. This refined model would offer a more accurate prediction of microbial growth in a broader range of conditions.

The two-step modelling approach, involving the use of two separate modelling stages, offers some disadvantages, depending on the problem at hand and the specific goals of the analysis [49]:➢**Dependency:** The success of the second-stage model heavily relies on the accuracy and reliability of the outputs from the first stage. Errors or uncertainties introduced in the initial stage can propagate to subsequent stages [50].➢**Resource Intensive:** Developing and running two models instead of one can require more time, computational resources, and expertise. This may not be feasible in cases with limited resources.➢**Limited Coverage:** The initial stage might focus on a subset of variables or simplified relationships, potentially missing out on important aspects of the problem. This could limit the accuracy and comprehensiveness of the final analysis.➢**Reduced Simplicity:** While the approach aims to tackle complexity, it can inadvertently lead to additional complexities due to the interaction between different models and stages.

## 4. One-Step Modelling Approach

The one-step modelling approach, also known as a single-stage modelling approach, involves using a single comprehensive model to analyse a complex problem or dataset [51]. Instead of breaking down the analysis into multiple stages, as performed in the two-step approach, all the relevant variables and relationships are incorporated into a single model. This approach has its own set of advantages, depending on the nature of the problem and the goals of the analysis [52].

➢**Holistic Understanding:** A one-step model offers a holistic view of the problem, allowing for the exploration of complex interactions and relationships among variables in a single framework.➢**Simplicity in Execution:** With a single model, there is no need to manage multiple stages or integrate outputs from different models. This can simplify the execution and interpretation of the analysis [53].➢**Integrated Insights:** All insights and predictions are generated within a single model, providing a unified output that does not require further integration or consideration.➢**Reduced Propagation of Errors:** Since there is no dependency on outputs from a previous stage, the potential for error propagation is reduced compared to multi-stage approaches [54].

The two-step approach helps to study different parts of a problem step by step, but it can spread mistakes, needs more resources, and might not show the full picture. On the other hand, the one-step approach looks at the whole problem at once, which makes it easier to use and reduces mistakes [49,54].

Recent findings have confirmed that the one-step fitting method is more accurate than the traditional two-step method, especially for extreme environmental conditions. It provides better interpretations and estimates of important parameters and is more efficient with smaller datasets. Although the two-step approach can be helpful in early stages, the one-step approach is preferable for a detailed analysis and precise results [49,54].

## 5. Machine Learning Modelling Approach

Machine learning (ML) approaches have become increasingly popular in the field of predictive food microbiology. These techniques leverage data-driven algorithms to develop models that can predict the microbial growth, spoilage, and safety of food products. ML methods offer the potential to capture the complex relationships between the various factors influencing microbial behaviour, leading to more accurate predictions and enhanced food safety. Here is an overview of how machine learning is applied in predictive food microbiology [55,56,57,58]: **Data Collection and Preprocessing:** Machine learning models require substantial amounts of relevant data. In predictive food microbiology, these data include information about factors such as temperature, pH, water activity, nutrient content, and more. Data preprocessing involves cleaning, transforming, and normalising the data to ensure their quality and suitability for modelling.**Feature Selection:** Feature selection involves identifying the most relevant variables (features) that influence microbial growth. Not all factors may be equally significant, and ML algorithms help in determining which features contribute most to the model’s predictive accuracy.**Model Selection:** There are various machine learning algorithms available, each with their strengths and weaknesses. Commonly used algorithms include decision trees, random forests, support vector machines, k-nearest neighbours, and neural networks. The choice of algorithm depends on the complexity of the problem and the nature of the data.**Model Training:** The selected ML algorithm is trained on the prepared dataset. During this training, the algorithm learns the relationships between the input features (e.g., temperature and pH) and the output (microbial growth). The goal is to minimise the difference between the predicted microbial growth and the actual observed data.**Model Validation and Evaluation:** Once the model is trained, it is essential to validate its performance on unseen data. This helps to ensure that the model can be generalised well to new situations. Common evaluation metrics include accuracy, precision, recall, F1-score, and area under the ROC curve (AUC-ROC).

Machine learning models developed using this approach can be used to predict microbial growth and behaviour under different conditions. For example, a model might predict the growth of a specific microorganism in a particular food product given various combinations of temperature, pH, and other factors. There are numerous advantages of machine learning in predictive food microbiology:➢**Complex Relationships:** ML algorithms can capture the intricate relationships between the multiple factors affecting microbial growth that might be difficult to model using traditional methods.➢**Flexibility:** Machine learning models can adapt to different types of data and are capable of handling non-linear relationships.➢**Data-Driven:** ML models can uncover patterns and insights in large datasets that might not be immediately apparent through a manual analysis.➢**Improved Accuracy:** The predictive accuracy of ML models can be higher compared to conventional models, as they can learn from diverse and extensive datasets.➢**Automation:** Once trained, ML models can automate predictions, allowing for real-time decision making in food production and safety management.

Machine learning approaches have the potential to revolutionise predictive food microbiology by providing accurate and adaptable models that predict microbial growth and behaviour [59,60]. However, their success depends on the availability of high-quality data, appropriate model selection, and continuous validation to ensure reliable predictions for food safety and quality management.

## 6. Comparison of the Machine Learning Modelling Approach to the Traditional Modelling Approach

Machine learning and traditional modelling techniques differ in their approaches to predicting the behaviour of microorganisms and estimating the shelf life of food products [61]. While traditional techniques rely on established mathematical models and computational methods, machine learning utilises algorithms to identify patterns and make predictions based on data [62]. Machine learning can handle complex and nonlinear relationships more effectively, allowing for analyses of large and diverse datasets. However, it often requires substantial amounts of data for training and may lack the interpretability of traditional models [63]. On the other hand, traditional modelling techniques offer more straightforward interpretations and are often based on well-understood biological and chemical principles. They may be more suitable for scenarios with limited data or when interpretability is crucial.

## 7. Shelf Life Prediction with Two-Step Modelling Approach

Predictive models have been employed to elucidate the growth dynamics of spoilage microorganisms, with a particular emphasis on the pivotal parameter of time required to attain a designated threshold level under fluctuating temperature conditions. Notably, the growth patterns of *Pseudomonas* spp., a prevalent microorganism frequently encountered across diverse food sources, have been meticulously examined and modelled through the application of various predictive frameworks.

Koutsoumanis [64] employed a logistic model to delineate the growth kinetics of *Pseudomonas* spp. in fish, stored over a temperature range spanning from 0 to 15 °C. Gospavic et al. [65] applied modified Gompertz and Baranyi models to construct a growth model facilitating the estimation of *Pseudomonas* spp. proliferation in poultry subjected to varying temperature conditions. Zhang et al. [66] used the Baranyi model to fit the number of *Pseudomonas* spp. in beef stored between 0 and 20 °C. Bruckner et al. [67] employed a modified Gompertz model to characterise the growth patterns of *Pseudomonas* spp. in pork and poultry meat, confined within the temperature range from 2 to 15 °C. Dabadé et al. [68] adeptly employed the Baranyi and modified Gompertz models to portray the growth dynamics of *Pseudomonas* spp. in tropical fresh shrimp, encompassing temperatures ranging from 0 to 28 °C. Lytou et al. [69] used the Baranyi model to describe the growth of total viable bacteria including *Pseudomonas* spp. in marinated and unmarinated chicken breast fillets stored at 4, 10, and 15 °C, and correlated these data with shelf life. Wang et al. [70] used a modified Gompertz model as their primary model to study the growth behaviour of *Pseudomonas* spp. on fresh mushroom under isothermal conditions. The aim of the study conducted by Tsironi et al. [71] was to develop and test the applicability of predictive models for a shelf life estimation of ready-to-eat (RTE) fresh cut salads in realistic distribution temperature conditions in the food supply chain. Tarlak et al. [72] used the modified Gompertz, logistic, and Baranyi models to describe the microbial growth data of *Pseudomonas* spp. on sliced mushroom (*Agaricus bisporus*) at different storage temperatures. They found that the Baranyi model gave better a goodness of fit for describing the growth behaviour of *Pseudomonas* spp. on sliced mushrooms than the modified Gompertz and logistic models. The growth behaviour of *Pseudomonas* spp. on button mushrooms under isothermal storage temperatures between 4 and 28 °C was described by the Baranyi model. The µ_max_ values obtained from the Baranyi model were correlated with the temperature using the Ratkowsky and Arrhenius models [73].

## 8. Shelf Life Prediction with One-Step Modelling Approach

The investigation conducted by Manthou et al. [74] encompassed an evaluation of the growth kinetics of naturally occurring *Pseudomonas* spp. on oyster mushrooms (*Pleurotus ostreatus*) during storage under varying isothermal conditions (4, 10, and 16 °C). A one-step modelling approach was utilised to quantitatively describe this behaviour. Within this framework, the Baranyi model was employed to estimate growth kinetic parameters such as maximum specific growth rate and lag phase duration, while a secondary square-root type model captured the impact of temperature on μ_max_. The overall fitness of the global model was evaluated through the root mean square error and the adjusted coefficient of determination, yielding values of 0.206 and 0.948, respectively.

The primary goal of this study was to propose an alternative approach, the one-step modelling approach, for the assessment and prediction of mushroom spoilage, particularly considering the presence of *Pseudomonas* spp. in the natural microflora of button mushrooms (*Agaricus bisporus*) stored across a temperature range spanning from 4 to 28 °C, as explained by Tarlak [75]. To achieve this, *Pseudomonas* spp. growth data were extracted from previously published curves for button mushrooms and subjected to simulation using both two-step and one-step modelling approaches. Various primary models, including the modified Gompertz, logistic, Baranyi, and Huang models, were implemented in these approaches to predict *Pseudomonas* spp. counts, integrating time and storage temperature. The relative performance of the two-step and one-step modelling approaches, as well as their employed primary models, were assessed in terms of goodness-of-fit indices, with the optimal modelling approach being determined based on these metrics.

In a similar manner, Tarlak and Khosravi-Darani [76] aimed to scrutinise and simulate the influence of storage temperature on the spoilage of aerobically stored chicken meat, employing both two-step and one-step modelling approaches. In this pursuit, diverse primary models were evaluated to assess their fitting capability for total bacterial counts in aerobically stored chicken meat. The one-step modelling approach demonstrated a notable enhancement in fitting capacity, irrespective of the selected primary model. Statistical indices, including the bias factor and accuracy factor, highlighted the superior predictive performance of the one-step modelling approach, particularly when coupled with the Huang model for predicting maximum specific bacterial growth rate values. Furthermore, the Huang model’s predictive prowess was evaluated under various non-isothermal storage conditions, exhibiting satisfactory statistical indices within the specified range. The validated one-step modelling approach emerged as a robust prediction tool for ascertaining chicken meat spoilage, enabling the prediction of shelf life as a function of storage temperature. Notably, the shelf life of chicken meat exhibited a decrement from 58 h to 16 h with an increase in storage temperature from 4 °C to 15 °C.

In a concurrent exploration, Tarlak and Pérez-Rodríguez [77] embarked on examining the effect of storage temperature on aerobically stored chicken meat spoilage, employing both two-step and one-step modelling approaches and incorporating diverse primary models, including the modified Gompertz, logistic, Baranyi, and Huang models. The study involved the collation of growth data points for *Pseudomonas* spp. from published studies conducted on aerobically stored chicken meat products. Temperature-dependent kinetic parameters, namely the maximum specific growth rate and lag phase duration, were characterised as a function of storage temperature using the Ratkowsky model within the framework of different primary models. The comparative evaluation of the modelling approaches encompassed the root mean square error, adjusted coefficient of determination, and corrected Akaike information criterion as the criteria for assessing fitting capability. The one-step modelling approach consistently exhibited a marked improvement in fitting performance, regardless of the chosen primary model. The models derived from the one-step modelling approach were subsequently validated against maximum growth rate data sourced from the independent published literature, where the Baranyi model demonstrated the optimal predictive capability, with both the bias and accuracy factors closely approximating 1. Ultimately, the shelf life of chicken meat, as influenced by storage temperature, was effectively predicted using both modelling approaches when employing the Baranyi model.

## 9. Shelf Life Prediction with Machine Learning Modelling Approach

The primary objective of the study conducted by Yildirim-Yalcin et al. [78] was to devise a predictive tool for forecasting the proliferation of total mesophilic bacteria in spinach through the application of machine-learning-based regression models, namely support vector regression, decision tree regression, and Gaussian process regression. The effectiveness of these models was subsequently juxtaposed against conventionally employed models, encompassing modified Gompertz, Baranyi, and Huang models, with the evaluation anchored in statistical metrics such as the coefficient of determination and root mean square error. The findings distinctly demonstrated that the machine-learning-based regression models yielded enhanced the predictive precision, showcasing a minimum R^2^ of 0.960 and a maximum RMSE of 0.154. This discerned accuracy underscores their viability as a credible alternative to conventional methodologies in the realm of predictive modelling for total mesophilic bacteria proliferation. Consequently, the software developed as a result of this study holds substantial promise as an alternative simulation approach within the domain of predictive food microbiology, capable of supplementing or even supplanting traditional methods.

The primary objective of the work performed by Yucel and Tarlak, [79] was to formulate distinct machine-learning-based regression methodologies, specifically decision tree regression (DTR), generalised additive model regression (GAMR), and random forest regression (RFR), for the anticipation of bacterial populations in beef. To achieve this goal, a dataset comprising 2654 bacterial data points pertaining to *Listeria monocytogenes*, *Escherichia coli*, and *Pseudomonas* spp., the most extensively investigated bacterial genera in beef, was procured from the ComBase database (www.combase.cc, accessed on 3 October 2023). The key predictor variables encompassed temperature, salt concentration, water activity, and acidity, pivotal in estimating the growth or survival behaviour of microorganisms within beef. Notably, the hyperparameters governing the proposed machine-learning-based regression methodologies were meticulously fine-tuned through a process of nested cross-validation. The efficacy of the proposed machine learning algorithms was appraised through the lens of their fitting capabilities, with statistical indices such as the coefficient of determination and root mean square error serving as evaluative metrics. Each of the applied regression techniques yielded commendable predictive outcomes, as evidenced by R^2^ values ranging from 0.931 to 0.949 and RMSE values spanning from 0.597 to 0.692, for individual microorganism populations. Among these, random forest regression (RFR) exhibited the most robust prediction capacity, prompting a more in-depth assessment of its efficacy. Upon external validation, the RFR model exhibited statistical indices within the range from 1.017 to 1.151 for the bias factor and from 1.137 to 1.370 for the accuracy factor, thereby affirming its reliability as an alternative means for simultaneously characterising the survival and growth behaviour of microorganisms in beef. Moreover, the RFR’s notable potential lay in its ability to circumvent the secondary model step intrinsic to the two-step modelling approach often utilised in predictive microbiology, thus presenting an efficient alternative simulation methodology in this field.

## 10. Conclusions

The realm of predictive food microbiology stands as a pivotal and vital domain within the broader field of food microbiology. With its intricate focus on forecasting and assessment, this field plays a significant role in enhancing our understanding of food shelf life dynamics. Irrespective of the primary model employed in the traditional approach, the utilisation of the one-step modelling approach substantially enhanced the predictive capacity of the models in quantitatively describing microbial counts. This improvement is attributed to the avoidance of error accumulation and propagation, which commonly occurs in two-step sequential nonlinear regression procedures. By connecting advanced methodologies, such as machine-learning-based regression models and comprehensive predictive approaches, predictive food microbiology provides invaluable insights into the intricate interplay between microorganisms and food products. Through its systematic evaluation of factors that influence microbial growth and behaviour, predictive food microbiology offers a comprehensive understanding of food product stability and safety over time. The insights gained from predictive food microbiology are indispensable for safeguarding consumer health, reducing food waste, and ensuring economic viability across the food industry. The ability to accurately estimate and extend the shelf life of food products not only contributes to consumer safety, but also has far-reaching implications for sustainability and resource management. As consumer demands continue to evolve and regulations become increasingly stringent, the predictive food microbiology field remains at the forefront of ensuring the quality and safety of our food supply. In essence, predictive food microbiology serves as a beacon of innovation and progress, bridging scientific rigor with practical applications. Its role in enhancing our ability to evaluate and manage food shelf life underscores its significance in shaping the future of the food industry. As technological advancements and interdisciplinary collaborations continue to propel this field forward, the continued exploration of predictive food microbiology promises to yield novel strategies for enhancing food preservation, minimising waste, and upholding the integrity of the global food supply.

## Data Availability

The data used to support the findings of this study can be made available by the corresponding author upon request.

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
