# Peer review of "The Use of Predictive Microbiology for the Prediction of the Shelf Life of Food Products"

_foods, 2023, doi:10.3390/foods12244461_

Round 1

Reviewer 1 Report

Comments and Suggestions for Authors

This review organizes information on predictive microbiology for prediction. It is a valuable study for reference. However, since it is a review article, the number of relevant references searched for in this article is really low. It should be supplemented and revised.

#1. As a review article, the number of relevant references searched for in this article is too low. It should be supplemented and revised.

#2. In Line 129-130, the important facultative aerobes are not included in the list of microorganisms.

#3. In Line 167-170, there are many other items that need to be evaluated in a risk assessment, so please gather relevant information and add to it with references.

#4. In the section 2.2 (Secondary models), it showed “to describe the impacts of many environmental conditions on the parameters of main models”. However, in the text, only the temperature-related part is discussed. Please gather relevant references and add to it.

#5. In the section 3 and 4 (Two-step modelling approach and One-step modelling approach), there is no citation of relevant reference at all. Please gather relevant reference and add to it.

#6. In Line 325-329, Machine learning modelling is an important method for predicting microbial growth in this review, and comparing the strengths and weaknesses of different methods is necessary.

#7. In the section 5 (Machine learning modelling approach), it is an important part of this review. It's too few to organize only one reference. Please gather relevant reference and add to it.

#8. In the section 6 (Shelf-life prediction with two-step modelling approach), it is not appropriate to organize only the information related to Pseudomonas spp. Please gather relevant reference and add to it.

#9. In the section 8 (Shelf-life prediction with machine learning modelling approach), It's too few references to organize. Please gather relevant reference and add to it.

#10 In Line 420-426, The one-step modelling approach is mentioned as a better approach, but this important point is not mentioned in the conclusion. It is suggested re-adding this section to the conclusion.

Comments on the Quality of English Language

Moderate editing of English language required

Reviewer 2 Report

Comments and Suggestions for Authors

There are certain points that need to kept in mind. These are as under

1.      Authors need to elaborate in detail the current state of microbial shelf-life prediction in the food industry, and how has predictive microbiology contributed to enhancing food safety and quality?

2.      The article lacks in elaborating key mathematical models and computational techniques used in predictive microbiology to estimate the shelf life of food products, and how do these models differ in their approaches?

3.      The authors lack in strengths and limitations of the two-step and one-step modelling approaches in predictive microbiology for estimating microbial shelf life in food products?

4.      The article lacks in explaining proper machine learning compare to traditional modeling techniques in predicting the behavior of microorganisms and the shelf life of food products. Please explain in detail. 

5.      Authors should also discuss that what are the implications of microbial contamination and spoilage for food safety, quality, and shelf life, and how do different predictive microbiology techniques address these concerns?

6.      How can the accuracy and reliability of microbial shelf-life predictions be enhanced.

7.      What are the potential risks associated with microbial contamination and spoilage of food products, and how do different predictive microbiology techniques aid in assessing and managing these risks?

8.      What are the implications of improved microbial shelf-life predictions for reducing food waste and enhancing consumer confidence in the safety and quality of food products?

These research questions can serve as a basis for exploring the various aspects of predictive microbiology and microbial shelf-life estimation, helping researchers and practitioners to further enhance food safety, quality, and consumer confidence while reducing food waste.
